# Creating synthetic spaces for higher-order topological sound transport

Hui Chen[1,4], Hongkuan Zhang [2,4], Qian Wu[1], Yu Huang[2], Huy Nguyen[1], Emil Prodan [3✉], Xiaoming Zhou [2✉] & Guoliang Huang [1✉]

Modern technological advances allow for the study of systems with additional synthetic dimensions. Higher-order topological insulators in topological states of matters have been pursued in lower physical dimensions by exploiting synthetic dimensions with phase transitions. While synthetic dimensions can be rendered in the photonics and cold atomic gases, little to no work has been succeeded in acoustics because acoustic wave-guides cannot be weakly coupled in a continuous fashion. Here, we formulate the theoretical principles and manufacture acoustic crystals composed of arrays of acoustic cavities strongly coupled through modulated channels to evidence one-dimensional (1D) and two-dimensional (2D) dynamic topological pumpings. In particular, the higher-order topological edge-bulk-edge and corner-bulk-corner transport are physically illustrated in finite-sized acoustic structures. We delineate the generated 2D and four-dimensional (4D) quantum Hall effects by calculating first and second Chern numbers and physically demonstrate robustness against the geometrical imperfections. Synthetic dimensions could provide a powerful way for acoustic topological wave steering and open up a platform to explore any continuous orbit in higher-order topological matter in dimensions four and higher.

[1] Department of Mechanical and Aerospace Engineering, University of Missouri, Columbia, MO, USA. [2] School of Aerospace Engineering, Beijing Institute of Technology, Beijing, China. [3] Department of Physics, Yeshiva University, New York, NY, USA. [4]These authors contributed equally: Hui Chen, Hongkuan Zhang. ✉email: prodan@yu.edu; zhxming@bit.edu.cn; huangg@missouri.edu

The physics of the integer quantum Hall effect (IQHE)[1] can manifest intrinsically in a condensed matter system[2–4] and the same physics can be emulated with classical degrees of freedom, such as electro-magnetic[5–8], mechanical[9–12], and acoustic[13–17]. Hall physics can be theoretically generalized to higher dimensions[18] and experimental realizations of the effect in four-dimensional (4D) were recently reported with atomic and photonic systems as well as electric circuits[19–21]. These experimental works point to two different strategies for emulating higher space dimensions in physical space. The first approach[21] uses the higher dimensional lattice coordinates and orbital indices only as labels for resonators rendered in physical space, in order to connect them based on a standard higher dimensional model[22]. The second approach relies on synthetic dimensions generated via space modulations[19,20], where the phases of the modulated structures can be used as adiabatic parameters that augment the physical space[23]. In essence, these phase variations can be treated as an additional global degree of freedom, usually called the phason, which lives on a torus. Even though the time-reversal symmetry is not broken for the static configurations of these systems, the IQHE physics emerges once the phason is pumped[24]. As such, this second approach is appealing because it does not require any active materials or other mechanisms to break the time-reversal symmetry.

The synthetic dimensions generated by space modulations can be explored statically, i.e., one point at a time[8,25–31], which is usually achieved by manual reconfiguration of the system. The phason space can be also explored dynamically and here the most sought application is where the configuration of the system is modified cyclically in time such that the original dynamical pumping proposed by Thouless[32] can be observed. The challenge is to cycle fast enough to overcome the dissipation of a signal that self-oscillates as it is pumped from one edge to the other. This is very challenging and it has been only recently achieved experimentally for one-dimensional (1D) but not for two-dimensional (2D) pumping[33–39]. A third strategy is to weakly couple modulated waveguides, whose effective dynamics are described by a Schrödinger-like equation, with the coordinate along the waveguides playing the role of time[23]. This strategy has been experimentally implemented to produce topological pumping with photonic and elastic degrees of freedom[20,23,40–43]. The last strategy mentioned above is not feasible with sound because acoustic wave-channels cannot be weakly coupled in a continuous fashion along with the guides and, to our knowledge, it has been never implemented with acoustic degrees of freedom.

In this work, we present a distinct strategy that functions in the opposite regime, where the waveguides are replaced by chains of coupled discrete resonators and strong couplings and modulations are also established in the transversal direction. In fact, the strategy we are proposing here can be better described as horizontal acoustic crystals carrying different phason values that are stacked and coupled with each other. By slowly varying the phason along the stacking direction, we demonstrate here that, with such an approach, we can explore any continuous orbit inside the phason space, and even control the speed along this path to control the shape of the pumped pattern. As a result, we can render these abstract trajectories, occurring in the synthetic dimensions, on the physical dimension along the stackings. In turn, this enables us to control the propagation of the acoustic modes in space as well as the temporal phases of the signals. In contrast to the waveguides experiments[20,23], where the samples had to be sliced for data acquisition, our designs enable a "non-demolition" measurement procedure that does not disrupt or alter the wave propagation.

With the control over the phason, we demonstrate edge-to-edge topological pumping of sound in 1D-modulated acoustic crystals, as well as edge-to-edge and corner-to-corner topological pumping in 2D modulated acoustic crystals. The higher-order topological corner-to-corner modes in our system are principally different from conventional realizations. We demonstrate these types of pumping processes that the topological sound transport is robust against random fluctuations in the resonator couplings. We also demonstrate that the pumping along a given orbit in the phason space occurs only in specific space directions. We delineate the generated 2D and 4D quantum Hall systems by calculating first and second Chern numbers. We also discuss various ways in which we can control these pumping processes and, moreover, we discuss topological mode steerings in 2D modulated acoustic crystals that are entirely specific to the 4D IQHE physics. We believe that our work breaks ground for engineering applications, where the couplings in an acoustic crystal can be programmed for selective and robust point-to-point distribution of acoustic signals.

## Results

**Physical rendering of synthetic spaces.** We describe here the mechanism behind the physical rendering of a synthetic space and we start by describing the general setting. It consists of a generic acoustic structure of discrete resonators such that each resonator has an address $(\mathbf{n}, m)$, with $\mathbf{n}$ being a horizontal label and $m$ a vertical one. The couplings in the horizontal plane occur through channels whose modulated widths are specified by a phason $\phi$, which lives on a phason space such as the $d_s$-torus. Here, $d_s$ represents the dimension of the synthetic space. The couplings in the vertical direction are established through uniform channels. Now, we assume an infinitely long system in the $z$-direction and consider any curve $\phi(z)$ in the phason space with bounded first derivative $\phi'(z)$, which we parametrize by the continuous $z$-coordinate. We use this curve to specify the phason values for each horizontal layer, specifically, $\phi_m = \phi(\epsilon z_m)$, where $z_m = m a_z$ is the physical coordinate of the $m$th layer. The parameter $\epsilon$ is small and is there to ensure that the variations of the phason from one layer to another are small. Compared with the wave-guide setting, the major differences are the discrete character of the $z$-coordinate and the strong coupling in the horizontal direction.

The resulting dynamical matrix $D_\phi$ governing the collective resonant modes depends on the chosen path $\phi(z)$ and, while $D_\phi$ is not periodic in the $z$-direction, it displays the following covariance property

$$T_\nu^\dagger D_\phi T_\nu = D_{\phi \circ \tau}, \tau(z) = z + \epsilon a_z, \qquad (1)$$

where $T_\nu$ is the vertical translation by $a_z$. If $Q_{\mathbf{n},m}$ is the acoustic resonant mode supported by the $(\mathbf{n}, m)$-cavity, then the pressure field of the resonant collective modes can always be sought in the form $\int dk_z \Psi_{k_z}(\mathbf{r}; \phi)$ (See Supplementary Note 1 for multimode expansions), with

$$\Psi_{k_z}(\mathbf{r}; \phi) = \sum_{\mathbf{n},m} e^{i k_z m} \varphi_{\mathbf{n},m}(\phi, k_z) Q_{\mathbf{n},m}(\mathbf{r}), \qquad (2)$$

and the covariance property (1) requires

$$\varphi_{\mathbf{n},m+1}(\phi, k_z) = \varphi_{\mathbf{n},m}(\phi \circ \tau, k_z) \qquad (3)$$

Since $\phi(z)$ is a smooth path, then, $\phi \circ \tau \approx \phi + \epsilon a_z \phi'$ and, as such, $\varphi_{\mathbf{n},m+1}(\phi, k_z) \approx \phi_{\mathbf{n},m}(\phi, k_z)$ to 0th order in $\epsilon$. In these conditions, the vertical dispersion of the $\varphi$-coefficients can be ignored and the $\varphi_{\mathbf{n},m}$ coefficients for a fixed $m$ become the eigen-

modes of the reduced Hamiltonian (see Supplementary Note 1)

$$H_{k_z}(\phi_m) = \sum_{\mathbf{n}} \nu(k_z)^2 |\mathbf{n}\rangle\langle\mathbf{n}|$$
$$+ \sum_{\langle\mathbf{n},\mathbf{n}'\rangle} \kappa_{\mathbf{n},\mathbf{n}'}(\phi_m)|\mathbf{n}\rangle\langle\mathbf{n}'|, \tag{4}$$

where the last sum goes over the neighboring cavities and $\kappa_{\mathbf{n},\mathbf{n}'}$ are the horizontal couplings, determined entirely by the phason value $\phi_m = \phi(\epsilon z_m)$. Also, $\nu(k_z)$ is the dispersion of the decoupled vertical channels and, if $\epsilon(\phi_m)$ is the eigenvalue of the $\varphi$-mode, then the value of $k_z$ at layer $m$ is determined by the relation $f^2 - \nu(k_z)^2 = \epsilon(\phi_m)$. The conclusion is that, by examining the horizontal spatial profiles of the collective resonant modes, one layer at a time, we can visualize the states of the Hamiltonian $H(\phi)$ along arbitrary paths inside the phason space.

So far we have established that the coefficients $\varphi_{\mathbf{n},m}$ for fixed $m$ are eigen-modes of Hamiltonian (4), but what are the weight and the phase of these modes in Eq. (2)? An expansion in the first order in $\epsilon$ (see Supplementary Note 1) reveals that the answer is supplied by the equation

$$i\epsilon\Gamma(\phi)\partial_\phi|\varphi(\phi)\rangle = [H(\phi) - \epsilon(\phi)]|\varphi(\phi)\rangle, \tag{5}$$

which governs the evolution of the $\varphi$-coefficients along the $\phi$-trajectory. The function $\Gamma(\phi)$ is determined entirely by the vertical mode dispersion. After a proper change of variable, Eq. (5) becomes the classic equation of adiabatic evolution[44], hence the amplitudes of the modes along the $z$-direction are all equal but a non-trivial phase $e^{i\alpha(z)}$ does develop along the stacking direction. It can be computed as the Berry phase of the Wilczek-Zee connection[45] along the path $\phi(z)$ The important conclusion is that we not only can control the spatial profiles of the modes but also their phases. The latter has been already proposed as vehicles for certain forms of information processing with classical meta-materials[46]. Last, it is worth mentioning that if the acoustic crystal is finite in the $z$-direction, the collective resonant modes are given by linear superpositions of the $k_z$ solutions of Eq. (5) and they occur at quantized values of the wavenumbers for which the top and bottom boundary conditions are simultaneously satisfied. Since the effective Hamiltonian from Eq. (4) is independent of the sign of the wavenumber, the main conclusion regarding the horizontal spatial profiles of the modes holds without modifications.

**1D topological pumping.** Figure 1a, b show a planar array of acoustic cavities coupled horizontally and vertically through channels. Each cavity has an address $(n, m) \in \mathbb{Z}^2$ and the thickness of the horizontal channel connecting $(n, m)$ and $(n + 1, m)$ resonators is modulated according to the protocol $h_{nm}^x = h_0[1 + \delta\cos(b_{n\bmod 3} + \phi_m)]$, where $h_0$ is the average thickness of the horizontal channels, $\delta$ is the modulation amplitude, and $b_j$'s are free parameters. The values of phason for each layer are set by $\phi(z) = \phi_i + (\phi_f - \phi_i)\frac{z}{L_z}$, where $\phi_i = -0.2\pi$, $\phi_f = 0.2\pi$ and $L_z = 16a_z$. This results in a variation $\triangle\phi = 0.026\pi$ from one layer to another, hence within the adiabatic conditions (see Supplementary Note 2 for further details).

By design, when $b_j = (j - 1)\frac{2\pi}{3}$, the effective Hamiltonian $H_{k_z}(\phi)$ in Eq. (4) is just the 1D Aubry-André-Harper model[47] associated with the 2D Hofstadter model at magnetic flux $\pi/3$, with $\phi$ playing the role of a quasi-momentum. Figure 1c shows the resonant spectrum of $H_{k_z}(\phi)$ as function of $\phi$ and $k_z$. The computation is carried out with COMSOL Multiphysics on the domain of a finite horizontal stack with $k_z$-twisted Bloch boundary conditions in the vertical direction. The bulk and the boundary spectra are shown with distinctive colors and, as expected from the Hofstadter butterfly[48], two bulk spectral gaps

are observed. Also from the Hofstadter butterfly, one can read that the lower and upper band gaps carry first Chern numbers $C_1 = \{-1, 1\}$. This is confirmed in Supplementary Note 1 by direct calculations of the Chern numbers for an entire phase diagram computed for various values of $b_j$'s. At last, as expected from the bulk-boundary correspondence for the 2D IQHE, topological edge modes are observed in the spectrum reported in Fig. 1c (see the blue sheets). At fixed $k_z$, there are precisely one chiral edge band per edge and the slopes of these bands are consistent with the values of the Chern numbers (see Supplementary Note 1).

We now focus on the spatial profiles of the modes, as excited at frequency $f = 4960$ Hz, indicated by the red horizontal sheet in Fig. 1c. It is chosen to intersect the dispersion surface of the edge modes such that we can visualize a topological pumping of sound. All modes along the curve resulted from the intersection of the $f = 4960$ Hz plane and the dispersion surfaces will be excited. As argued in the previous section, $\phi$ is resolved by the $z$-coordinate, hence this pumping curve, shown again in Fig. 1d, can be parametrized by the physical coordinate along the stacking, $(\phi(z), k_z(z))$. In other words, the spectral data from Fig. 1d have been rendered in the physical dimension, for us to observe. We now can understand the spatial profiles of the excited modes, when examined one stack at a time. Along the horizontal stack at a given coordinate $z$, one should observe the mode of $H(\phi(z))$ at energy $f^2 - \nu(k_z(z))^2$. The evolution of this mode along the pumping curve is shown in Fig. 1d, which confirms that sound is indeed pumped from one edge to the other. For example, the left edge state denoted in the inset (1) is selected as an initial state with a negative pumping value, which remains localized on the left boundary with the adiabatic increase of the pumping parameter and the corresponding wavenumber. When the pumping parameter approaches $\phi = 0$, the left edge state becomes the bulk state as inset (2) depicted. As the pumping parameter increases further, the bulk state is then transformed into an edge state (3) localized at the right side. Then our prediction is that sound is transported from one side to the opposite side of the structure and this topological sound steering can be witnessed by walking along vertical coordinate. Experimental observation and confirmation of the adiabatic pumping via topologically protected boundary states are reported in Fig. 1e. Here, the sound is injected into the bottom-left corner of the structure and the pressure field is mapped by a microphone for each site (see Methods). As one can see, the pressure distribution indeed renders the topological pumping process in the physical dimensions. The experimental observation is also verified by the numerical simulation based on the exact geometry (Fig. 1f, see also Supplementary Movie 1 for 1D transient topological edge pumping). The minor difference between experiment and simulation may be attributed to manufacturing deviations from connecting adiabaticity and perfect coupling to edge states (Details of the sample manufacturing, numerical simulation, and experimental testing can be found in Methods). Let us also point out that there is a substantial region where the wave has a bulk character and where dissipation mostly occurs (Details of loss effects can be found in Supplementary Note 1). This region can be reduced by optimizing the function $\phi(z)$ from a linear to a tangent hyperbolic profile.

**2D topological pumping.** We now investigate the three-dimensional (3D) acoustic structure shown in Fig. 2a, b, engineered to have a phason $\phi = (\phi^x, \phi^y)$ living on 2-torus. In this case, each cavity has an address $(\mathbf{n}, m) \in \mathbb{Z}^3$ and the thicknesses of the horizontal connecting channels in the $\alpha = x, y$ directions are modulated according to the protocol

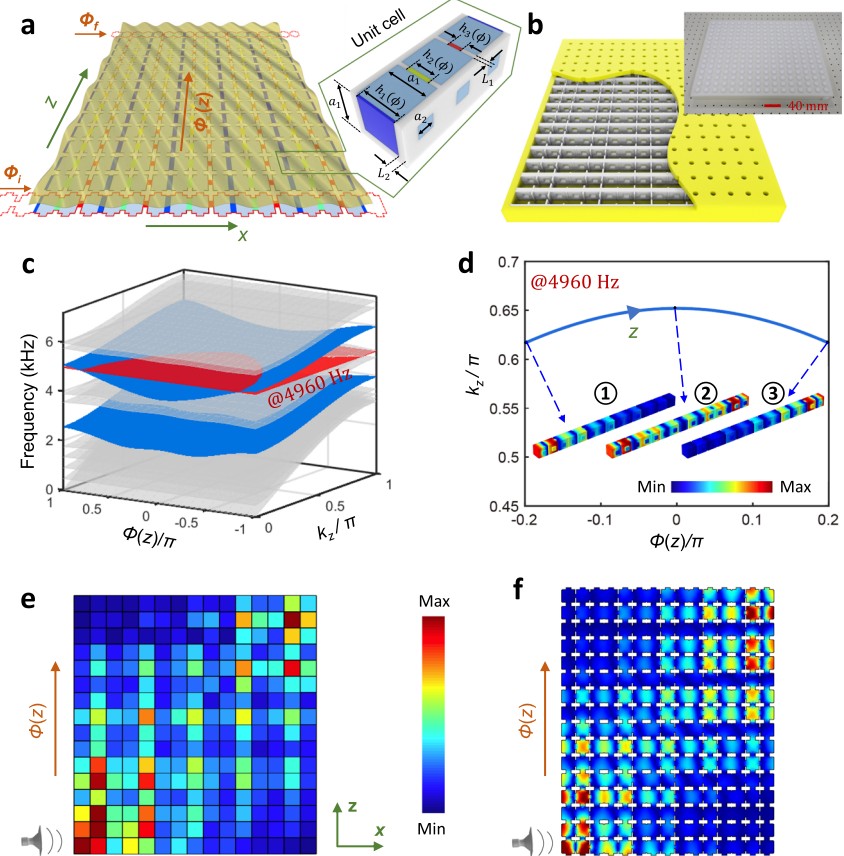

**Fig. 1 2D channel-modulated acoustic crystal and its dispersion property. a** Schematic of the 2D channel-modulated acoustic crystal and its unit-cell.
**b** Photograph of the printed 2D channel-modulated sample. **c** Dispersion diagram of a supercell composed of 15 coupled cavities, terminated by a hard-wall
boundary along the $x$-direction, and a periodic boundary condition along the $z$-direction. The topological pumping of edge states is represented by blue
surfaces, whereas the bulk bands are indicated by gray surfaces. **d** Wave number $k_z$ as a function of the pumping parameter for frequency $f = 4960$ Hz,
corresponding to the cross-section illustrated by the red horizontal sheet in **c**. The insets show the topological mode from left to right localization with the
change of the pumping parameter $\phi$ from $-0.2\pi$ to $0.2\pi$ along the $z$-direction. The color scale corresponds to the pressure amplitude. **e, f** Experimental **e**
and numerical **f** demonstrations of topological edge pumping in the 2D crystal with the frequency at $f = 4960$ Hz. The color scale corresponds to the
pressure amplitude.

$h^\alpha_{\mathbf{n},m} = h_0[1 + \delta\cos(b^\alpha_{n_\alpha \bmod 3} + \phi^\alpha_m)]$, while the vertical connect-
ing channels are uniform. By design, the effective horizontal
Hamiltonian (4) is just a sum of two copies of the Hamiltonian
from 1D topological pumping, $H(\boldsymbol{\phi}) = H(\phi^x) \otimes I + I \otimes H(\phi^y)$,
which is known to host 4D QHE physics[20].

The system can be pumped along an orbit inside the phason
space by using the strategy described in Section "Physical
rendering of synthetic spaces". The particular crystal is designed
to pump along the diagonal orbit $\boldsymbol{\phi}(z) = (\phi(z), \phi(z))$, with
$\phi(z) = \phi_i + (\phi_f - \phi_i)\frac{z}{L_z}$, where $L_z = 15a_z$. In Supplementary
Note 3, we present additional 3D acoustic structures, engineered
to pump along the orbit $(\phi(z), \phi_i)$. Figure 2c shows the dispersion
surfaces of the resulting effective Hamiltonian (4) as a function of
the pumping parameter $\phi$ and $k_z$. The computation, which is
carried as for the 2D structure, reveals two bulk gaps and, in
addition, two spectral sheets highlighted in blue for which the
modes are localized along two edges, as well as one sheet
highlighted in green for which the modes are localized at the
corners. In Supplementary Note 1, we demonstrate that both bulk
gaps carry non-trivial second Chern numbers. Figure 2d (upper
panel) shows the pumping curves resulted from the intersection
of the dispersion diagram with the plane at frequency
$f = 7498$ Hz, with the latter highlighted in red in Fig. 2c. In
contrast to the 1D topological pumping, there is more than one

such pumping curve. Similarly, Fig. 2d (lower panel) shows the
pumping curve resulted from the intersection with the plane at
frequency $f = 6175$ Hz, with the latter highlighted in purple in
Fig. 2c. All these pumping curves are parametrized by the $z$-
coordinate and, as in the 1D case, we can predict that, if one
examines the horizontal stack at coordinate $z$, one should observe
the eigen-modes of $H(\phi(z))$ at energy $f^2 - \nu(k_z(z))^2$. Samples of
these modes are rendered in the inset of Fig. 2d and, as one can
see, both pumping curves that we engineered are very special.
Indeed, as one pumps along the blue contours, the mode is
pumped from one pair of edges to the opposite pairs of edges,
whereas if one pumps along the green curve the mode is pumped
from one corner to the opposite corner. Both pumping processes
proceed through a bulk delocalization transition.

In Fig. 3a, we demonstrate experimentally that the predicted
edge-bulk-edge pumping is indeed rendered in the physical space
of the 3D structure. The acoustic wave is excited by a sound
speaker along the bottom edge at $f = 7498$ Hz and the pressure
distribution is measured by the microphone, layer by layer along
the stacking direction (see Methods). As seen in Fig. 3a, the
pressure field does evolve from the left to the right edge as
one walks along the stacking direction. It is interesting to note
that the edge states that we excite have the same energies as bulk
states, which are also seen as being excited in the very bottom

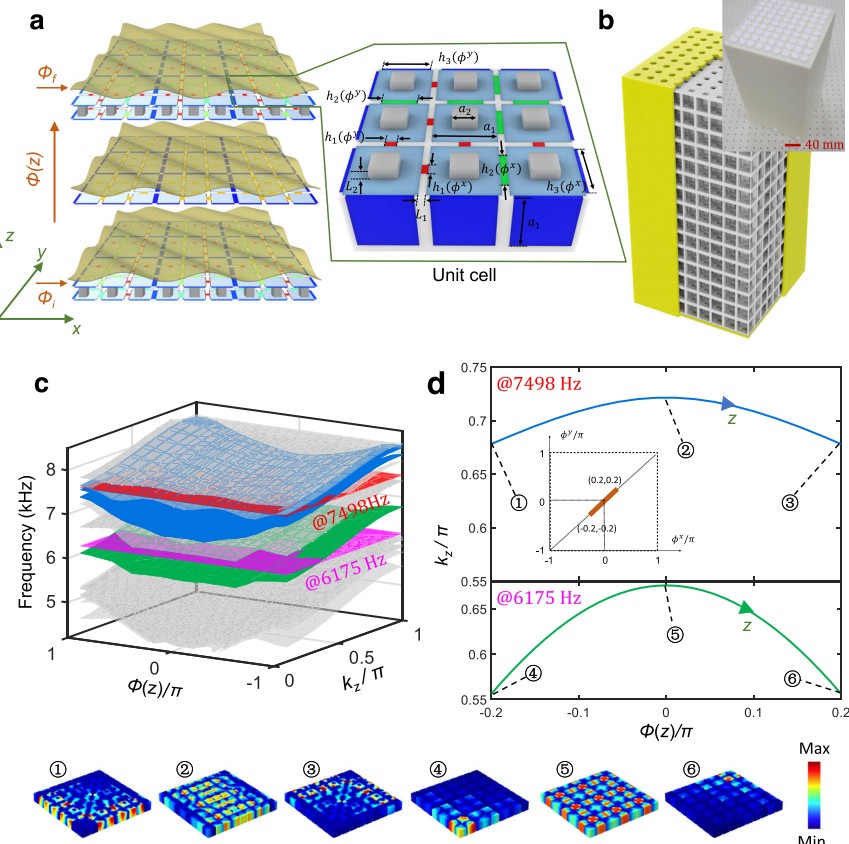

**Fig. 2 3D channel-modulated acoustic crystal and its dispersion property. a** Schematic of the 3D channel-modulated acoustic crystal and its unit cell. **b** Photograph of the printed 3D channel-modulated sample. **c** Dispersion diagram of a supercell composed of 9 × 9 coupled cavities, terminated by a hard-wall boundary along $x$ and $y$ directions, and a periodic boundary condition along the $z$-direction. Bulk modes are shown in gray, topological pumping of edge and corner modes are in blue and green, respectively. **d** Wave number $k_z$ as a function of the pumping parameter for fixed frequencies $f = 7498$ Hz (upper panel) and $f = 6175$ Hz (lower panel), corresponding to the cross-section illustrated by the red and purple planes in **c**. The inset in the upper panel shows the diagonal orbit with the phason value from $(-0.2\pi, -0.2\pi)$ to $(0.2\pi, 0.2\pi)$. The insets below **c** and **d** show representative mode shapes for each type of mode. The color scale corresponds to the pressure amplitude.

layer. However, as the bulk states extend throughout the horizontal layer, they experience increased dissipation and, as such, fade away along the stacking direction. On the other hand, the topologically pumped states are long-propagated, which further demonstrates the advantages of the topological steering of sound. A numerical simulation is also conducted to validate our experimental observation (Fig. 3b, the corresponding pressure distribution along $z$-direction can be found in Supplementary Figure 12, see also Supplementary Movie 2 for 2D transient topological edge pumping).

In Fig. 3c, we demonstrate experimentally that also the predicted corner-bulk-corner pumping can be rendered in the physical space of our 3D structure. Indeed, with the source placed at a bottom corner and with the frequency adjusted at $f = 6175$ Hz, one sees the measured pressure field evolving towards the opposite corner as one walks along the staking direction. This pumping is resolved much better than in the previous case, because the source couples less effectively to the bulk modes, hence the latter are not excited in this setup. Also, due to the reduced dimensionality of the mode, the dissipation is weaker and the pressure field can be sustained longer along the stacking direction. Numerical simulations for the entire 3D structure are reported in Fig. 3d and they validate our experimental findings (the corresponding pressure distribution along $z$-direction can be found in Supplementary Figure 12, see

also Supplementary Movie 3 for 2D transient topological corner pumping).

The emergence of the chiral boundary spectrum that makes possible the topological pumpings observed in Fig. 3 is owing to the second Chern number of the gaps, which is the strong topological invariant in 4D. As it is the case for any strong topological invariant, topological boundary spectrum emerges regardless of how the boundary of the crystal is cut, provided the available quasi momenta are properly sampled. In our case, the phason plays the role of synthetic momenta and this implies that the pumping process along a given phason orbit manifests only in a particular space direction. To demonstrate that the pumping processes are indeed highly directional, we simulate the acoustic characteristics of the 3D structure for different phason orbits and with the source placed at different space locations and encountered the following scenarios: (1) propagation along a facet; (2) propagation along an edge; (3) pumping in the $x$ but not in the $y$ direction; (4) pumping in the $y$ but not in the $x$ direction; (5) pumping along the first diagonal $x = y$ but not along the second diagonal $x = -y$. (6) pumping along the second diagonal $x = -y$ but not along the first diagonal (see Supplementary Note 3). We also demonstrate what we call an "antagonistic" effect, which manifests as follows: If edge-to-edge or corner-to-corner pumping is observed with the source placed on one edge or corner, respectively, then the pumping or any propagation is

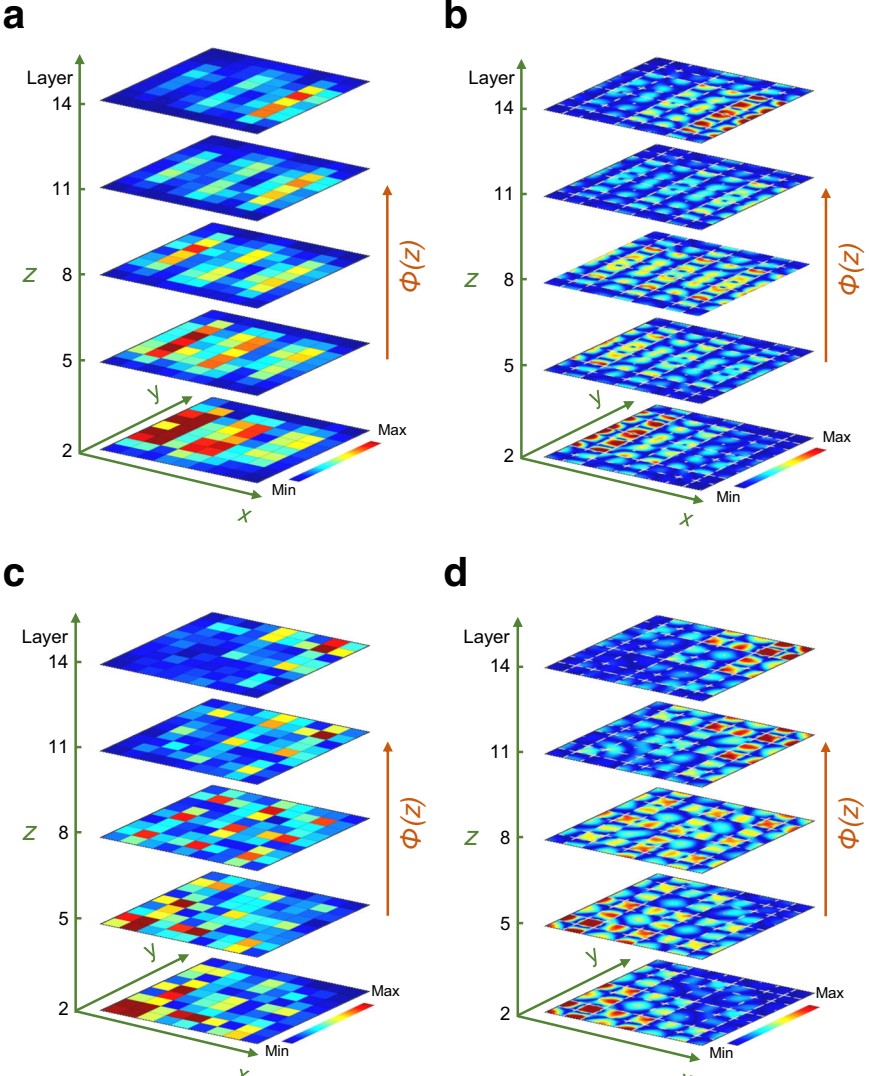

**Fig. 3 Topological pumpings in the 3D channel-modulated acoustic crystal. a, b** Experimental **a** and numerical **b** observation of edge-bulk-edge pumping. The sound is injected at the bottom-left edge with the frequency at $f = 7498$ Hz. **c, d** Experimental **c** and numerical **d** observation of corner-bulk-corner pumping. The sound is injected at the bottom-left corner with the frequency at $f = 6175$ Hz. Note that the pumping parameter $\phi(z)$ for each horizontal layer of these samples is evenly distributed from $-0.2\pi$ to $0.2\pi$ along the $z$-direction. The color scale corresponds to the pressure amplitude.

completely absent if the source is moved to the opposite edge or corner. In this respect, the structure acts like a perfect "transistor" because the pumping can be turned on and off by 180° rotations. In fact, by combining all the effects listed above, our 3D structure can be transformed into a multifunctional and programmable acoustic device for sound transport and distribution. We stress that our structure does not possess any crystalline symmetry so that the bulk polarizations are not quantized[49]. Therefore, the higher-order topological corner modes in our system are fundamentally different from previous realizations based on quantized quadrupole polarization[50–52] or quantized Wannier centers[53–56].

Topological phases with non-zero second Chern numbers display intriguing wave transport characteristics, such as robustness against impurities or defects[22]. It is of interest to quantify the extent of the topological protection in such conditions. To evaluate that, a 3D hollow structure is constructed by removing nine cavities at the center of the system and the topological edge-bulk-edge and corner-bulk-corner pumpings in the defected structure are experimentally measured, as shown in Fig. 4a, c, respectively. When comparing performance with the system

without defects (Fig. 3a, c), both edge and corner modes can be smoothly pumped despite the presence of defects on a relatively large scale. This confirms that the topological pumping is immune against back reflections from defects or discontinuity. The experimental observation is also verified by the numerical simulation based on the exact geometry (Fig. 4b, d). In addition, a robust topological pumping due to disorder in the pumping parameters is also numerically evaluated in the Supplementary Note 4.

## Discussion

We demonstrate a robust strategy to explore the global degrees of freedom of modulated wave media. While we have exemplified here only simple orbits inside the phason spaces, the method has no limitations on the geometry and topology of these orbits. For example, in the present study on the 2D pumping, we explored the fundamental loops $x$ and $y$ of the 2-torus (see the horizontal and vertical orbits in Supplementary Note 3) as well as the diagonal orbit, which is topologically equivalent to the combination $C_x + C_y$. The 2-torus, however, supports an infinite

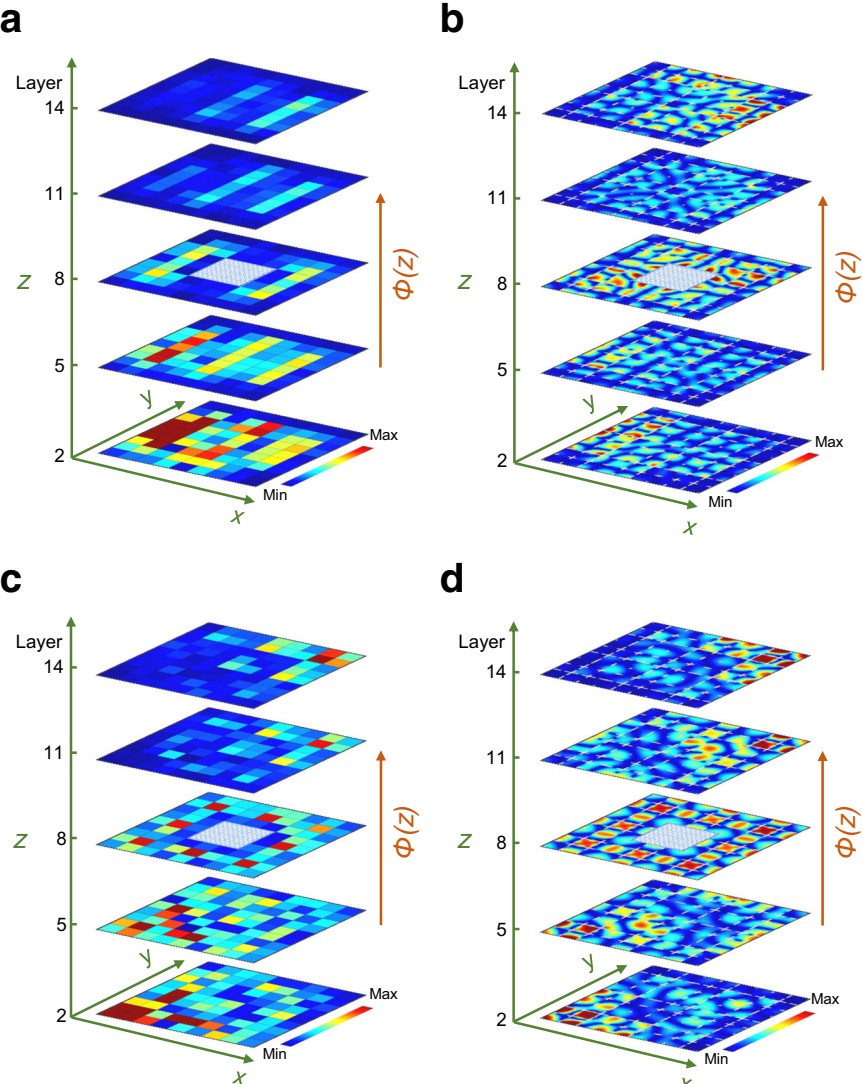

**Fig. 4 Robust topological pumpings in the 3D hollow acoustic crystal. a**, **b** Experimental **a** and numerical **b** observation of edge-bulk-edge pumping. The sound is injected at the bottom-left edge with the frequency at $f = 7498$ Hz. **c**, **d** Experimental **c** and numerical **d** observation of corner-bulk-corner pumping. The sound is injected at the bottom-left corner with the frequency at $f = 6175$ Hz. Note that the pumping parameter $\phi(z)$ for each horizontal layer of these samples is evenly distributed from $-0.2\pi$ to $0.2\pi$ along the $z$-direction and the hollow area is indicated by the blue grid plane where nine cavities are removed. The color scale corresponds to the pressure amplitude.

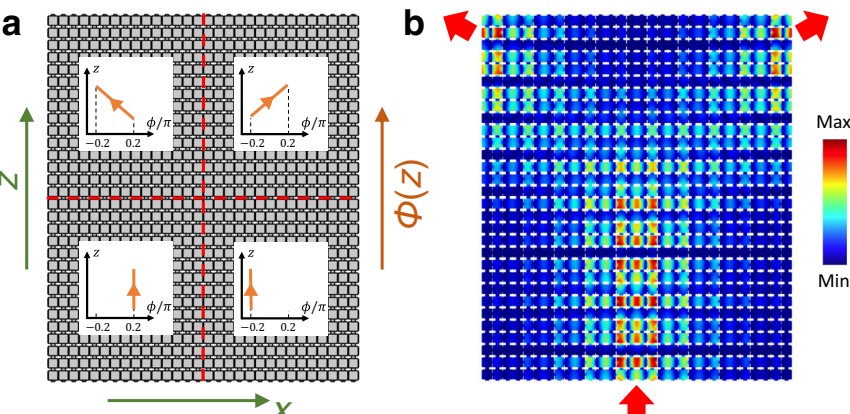

**Fig. 5 Topologically protected acoustic beam splitter. a** Schematic of the phason engineering in the 2D acoustic system. **b** The pressure field distribution of the beam splitter. The sound is injected at the center of the bottom edge with the frequency at $f = 4960$ Hz. The color scale corresponds to the pressure amplitude.

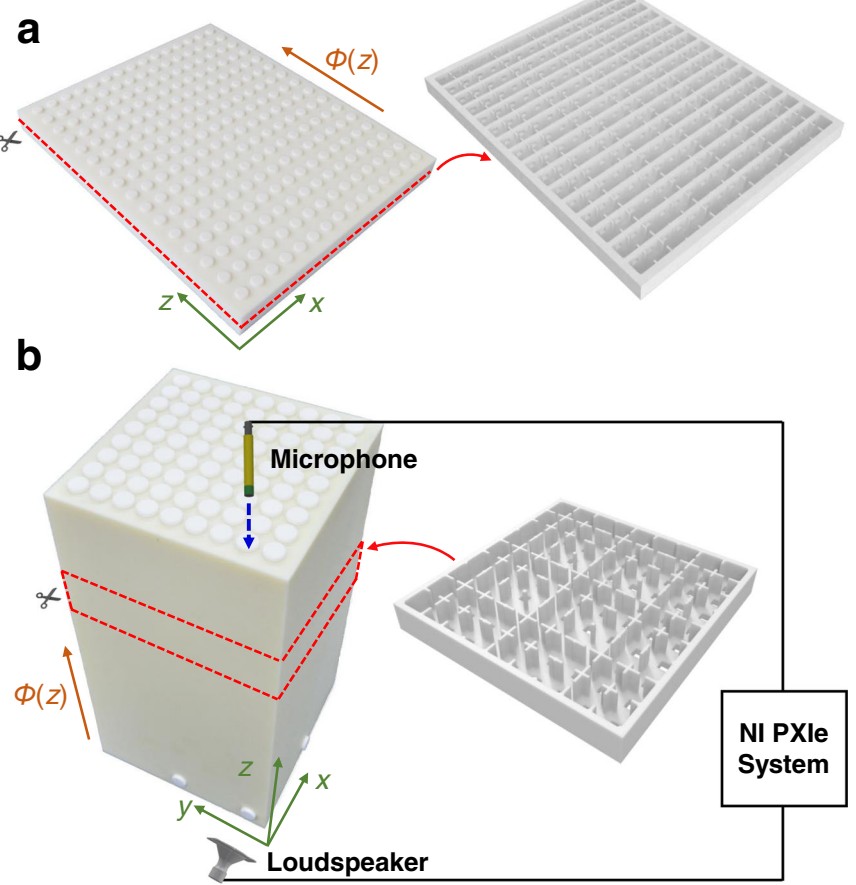

**Fig. 6 Fabricated samples and experimental setup. a** Photograph of the printed 2D channel-modulated sample. **b** Photograph of the printed 3D channel-modulated sample and schematic of the experimental setup. The microphone is inserted from the top of the sample to measure the sound pressure of each cavity.

number of topologically distinct paths, which can be in principle explored with the methods demonstrated in this work. It remains to be seen if the phases of the sound signals can be resolved as predicted in Section "Physical rendering of synthetic spaces", in which case the steering of the modes in both space and time domains could be controlled with the same device. In our opinion, the phason engineering may exhibit the possibility to implement the topological split-flow device, such as the topological beam splitter. In Fig. 5a, we create an acoustic beam splitter to engineer two-way beam splitting. In the current design, we consider a 2D system with different phason orbits in four quadrants: in the first quadrant, the phason value is linearly distributed from $-0.2\pi$ to $0.2\pi$ along the $z$-direction; in the second quadrant, the phason value is linearly distributed from $0.2\pi$ to $-0.2\pi$ along the $z$-direction; in the third quadrant, the phason value is held constant with $\phi = 0.2\pi$ along the $z$-direction; in the fourth quadrant, the phason value is held constant with $\phi = -0.2\pi$ along the $z$-direction. The input point is located at the bottom edge between the third and fourth quadrants. As shown in Fig. 5b, the sound stays confined to the interface until it arrives at the junction of four quadrants. Then, it splits into the first and second quadrants, and eventually reaches the two-end sites. Thanks to topological protection, the propagation is immune against back reflection from discontinuity. As such, our design, based on phason engineering and topological pumping, provides an avenue for the application of acoustic beam splitters. In addition, by replacing the waveguides with discrete coupled resonators, one now has the opportunity to engineer the

dispersion with respect to $k_z$ quasi-momentum. This will involve modulations along the vertical direction and this opens a dimension in the design space, which is yet to be explored.

In conclusion, we have evidenced the topological sound transport in modulated acoustic crystals through edge-to-edge topological and corner-to-corner topological pumpings associated with the 2D and 4D quantum Hall effects by the physical rendering of synthetic spaces. These observations imply that the system is characterized by a non-zero Chern number and therefore the topological pumping is immune to bulk scattering and exhibits strong protection against design imperfections. The modulated acoustic crystals with synthetic spaces offer a platform and route for efficient acoustic topological mode transport by engineering desired patterns on a phason-torus, and the higher dimensional quantum Hall effect may provide surface acoustic phenomena in the finite structure. The phason space augments the physical space and this opens a door to higher-dimensional physics in acoustics and mechanics. Although we focused on the acoustic implementation using synthetic spaces, our approach can be generalized to other degrees of freedom, such as additional frequency dimensions can also be harnessed for the frequency modulation[57,58]. Going forward, it will be important to develop and explore such broader connections, as the idea of topological matter in synthetic dimensions is very general and the extension of this approach to other complex orbits is much awaited. At last, we emphasize that, in order to achieve a reasonable adiabatic regime, the number of stacks in our experimental set-ups is appreciable and, whereas this is perfectly fine for demonstration

purposes, it will be an obstacle for practical applications. Recently, Fedorova et al.[59] showed that the non-adiabatic effects can be compensated using modulated dissipative channels and that such a strategy can be used to achieve quantized topological pumping with fast-driven cycles. It will be interesting to explore if this strategy can be deployed for our acoustic crystals in order to reduce the number of stacks needed for the topological pumping of sound.

## Methods

**Experimental specification.** The stereolithographic 3D printing technique is used to produce the experimental samples. The acoustic systems consisting of air cavities connected by modulated channels are made of photopolymer, which serves as acoustically hard walls owing to a high impedance mismatch compared with the air. The dissipation effects are minimized by employing the state of the art in 3D printing, which delivers acoustic crystals made of a single solid piece of high-quality polymer. Furthermore, the modulation of the connecting channels is adjusted until the topological gaps opened appreciably such that the dynamics of the topological pumped modes is virtually unaffected by dissipation effects, at least for the system sizes considered in our studies.

The fabricated 2D channel-modulated sample (Figs. 1b and 6a) consists of $15 \times 16$ cavities with thickness-modulated channels in the $x$-direction and thickness-constant channels in the $z$-direction. The thickness of modulated channels is $h_j = h_0 \left[ 1 + \delta \cos \left( b_j + \phi \right) \right] (j = 1, 2, 3)$, where $h_0 = 12$ mm, $\delta = 0.6$, and $\{ b_1, b_2, b_3 \} = \{ 0, 2\pi/3, 4\pi/3 \}$. The side length of the cubic cavities is $a_1 = 20$ mm, the length of the modulated channels is $L_1 = 2$ mm, the length of the vertical connecting channels coupling the layers is $L_2 = 3$ mm, and their side length is $a_2 = 8$ mm.

The fabricated 3D channel-modulated sample (Figs. 2b and 6b) consists of $9 \times 9 \times 15$ cavities with thickness-modulated channels in the $xy$-plane and thickness-constant channels in the $z$-direction. The thickness of modulated channels is $h_{\alpha j} = h_0 \left[ 1 + \delta \cos \left( b_j^\alpha + \phi^\alpha \right) \right] (\alpha \in \{ x, y \}, j = 1, 2, 3)$, where $h_0 = 11.2$ mm, $\delta = 0.75$, and $\{ b_1^x = b_1^y, b_2^x = b_2^y, b_3^x = b_3^y \} = \{ 0, 2\pi/3, 4\pi/3 \}$. The side length of the cubic cavities is $a_1 = 20$ mm and the length of the modulated channels is $L_1 = 2$ mm. The length of the vertical connecting channels coupling the modulated layers is $L_2 = 3$ mm, and their side length is $a_2 = 8$ mm. The pumping parameters $\phi(z)$ in these two samples are evenly distributed from $\phi_i = -0.2\pi$ to $\phi_f = 0.2\pi$ along the $z$-direction.

Figure 6b shows the experimental setup. A cylinder loudspeaker (diameter, 7 mm; height, 5 mm) and a microphone ($20 \times 9 \times 4$ mm) are used as the sound source and the acoustic pressure probe, respectively, both of which are small enough to be inserted in the cavities. The NI PXIe data acquisition system (PCI-4461 and PXIe-4610) is equipped for the experimental measurement, and a lab-made LabVIEW program controls the PCI-4461 to generate a sinusoidal signal that is amplified by PXIe-4610 to stimulate the speaker to emit sound waves of a given frequency. Meanwhile, the microphone is inserted from the top of the sample to measure the sound pressure signals at the center of each cavity and the data are recorded by the PCI-4461 data acquisition card. In each measurement, the unscanned holes are sealed by caps to avoid sound leakage.

**Numerical simulations.** The full-wave finite-element method simulations in this work are all performed using the commercial software COMSOL Multiphysics. The 3D geometry is implemented by filling with air (density $\rho = 1.225$ kg/m$^3$ and speed of sound $v = 343$ m/s). Eigenmode calculations within the "acoustic module" are carried out to find the dispersion relations of the supercell. For calculations of the relation between the wave number and pumping parameters as well as the pressure eigenfunctions of the supercell, the simulations are implemented by the "PDE (partial difference equation) Interfaces module" in which we write the coefficient form of the PDE with two independent variables: the density and the speed of air. Large-scale simulations are then implemented by the "acoustic module" and frequency domain calculations are performed to obtain the steady acoustic pressure fields. Supplementary Movies 1–3 are generated by the "acoustic module" and time-domain calculations are performed to obtain the transient acoustic pressure fields.

## Data availability

All the data supporting the findings of this study are available from the corresponding authors upon reasonable request. Source data are provided with this paper.

## Code availability

The computer code and algorithm that support the findings of this study are available from the corresponding author upon reasonable request. Source data are provided with this paper.

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

## Acknowledgements

This work is supported by the Air Force Office of Scientific Research under grant no. AF 9550-18-1-0342 and AF 9550-20-0279 with Program Manager Dr. Byung-Lip (Les) Lee, the NSF EFRI under grant no. 1641078, the NSF CMMI under Award no. 1930873 with Program Manager Dr. Nakhiah Goulbourne, the Army Research Office under Grant No. W911NF-18-1-0031 with Program Manager Dr. Daniel P. Cole. X.Z. acknowledges support from the National Natural Science Foundation of China (11872111, 11991030, 11991033, and 11622215) and 111 project (B16003). E.P. acknowledges financial support from the W.M. Keck Foundation and USA National Science Foundation through grant DMR-1823800.

## Author contributions

H.C. and G.H. conceived the concept; H.C. and E.P. performed theoretical investigations; H.C., Q.W., H.N., and E.P. performed numerical investigations; H.Z. and Y.H. conducted experiments; H.C., E.P., and G.H. analyzed methods and interpreted mechanisms; G.H. and X.Z. supervised the research. All the authors discussed the results and wrote the manuscript.

## Competing interests

The authors declare no competing interests.
