## [Peer Review File · Nature Communications]

REVIEWER COMMENTS

Reviewer #1 (Remarks to the Author):

In the manuscript titled "Creating synthetic spaces for higher-order topological sound transport", the authors realize higher-order topological boundary modes transportation by employing synthetic dimensions. More specifically, the synthetic dimension attributes to the existence of the topological boundary mode and varying the synthetic coordinate along z direction induces boundary modes transport, alike to the well-known Thouless pumping. The physics behind is solid, experimental results are also present to support the simulation results.

This work is interesting in sound wave manipulation, while before I get the conclusion to support its publication in Nature Communication, I would like the authors to address the following remarks and comments.

1) The author claims the system obeys Eq. (5) in the main text and said "hence the amplitudes of the modes along the z direction are all equal", which means there would be no reflection when wave propagates along z direction. While, as I can imagine, in extreme case when ϕ is fixed, since the system is periodic instead of be a straight waveguide along z direction, reflection induced by multiple scattering is expected. More simulation results to show the reflectivity from the system is appreciated. How would the reflection affect the outcome of the results?

2) This is a question related to the first one. I note that the total propagating distance along z direction is not long compared to the previous study (Ref [32]), while "adiabatic pumping" requires a slow variation of the parameters. It is important to show that the adiabatic condition is indeed satisfied in this setup. Can the authors provide a quantitative study to support the "adiabatic" claim?

3) Could you please figure out the pressure distribution along z direction in boundaries that wave localize?

3) The author shows the system is a tight-binding system and presents the fitting results in Supplemental Fig. S1. My puzzle is, since the system is a 3x3 model, why only two bands are presented? Why specific ϕ is selected?

4) Sound waves of 7 KHz in 3D printing structures may suffer considerable loss, which will broaden the spectrum, making the boundary mode and bulk mode overlap. Discussion about the loss is absent in the whole article.

5) Two bands with nonzero second Chern number are presented in Fig. S3b, what about other bands and what is the relation between second Chern number and the corner mode?

6) Color bars for the eigenstate field distributions in all figures are absent.

Reviewer #2 (Remarks to the Author):

In this work, 1D and 2D topological boundary pumping is demonstrated using arrays of coupled acoustic cavities. Effectively, linear sound propagation is controlled by a careful design of the array structure and coupling to topological boundary modes is done using a speaker [akin to previous works in the field: Nature Physics volume 15, pages357–361(2019)]. The chosen metamaterial structures and experimental ideas/observables reproduce the results from Refs. [23] and [20] for acoustic devices. The robustness of the 2D pumping to disorder in the bulk is further explored. The extension of topological boundary pumping is important to for potential applications. Connection to high-order topological insulators is commented on.

The exploration of topological effects in driven-dissipative metamaterials has started in the optics domain and since sparked immense activity in other linear media. The present results similarly provide a comprehensive transition of topological boundary pumping results from optics to acoustics. As such, they provide a novel platform for harnessing such effects for applications. I would, therefore, recommend publication in Nat. Commun. after the following points are discussed:

1. The topology of pumps relies on the adiabatic theorem. At the same time, in the experiment adiabaticity is broken. Nevertheless, boundary pumping is observed. I would like the authors to discuss this point, see, e.g., Nature Communications volume 11, 3758 (2020).

2. As the present study is mostly focused on transferring ideas from other fields for the purpose of novel applications, I would like the authors to provide more details on the proposed metrology applications.

Response to the Reviewers' Comments

The authors would like to take this opportunity to thank the reviewers for their insight and suggestions. The paper is improved based on the reviewers' comments. The relevant changes are highlighted in red in the revised manuscript and Supplementary Information. Below are detailed point-to-point replies to the reviewers' comments.

Reviewer #1 (Remarks to the Author):

In the manuscript titled “Creating synthetic spaces for higher-order topological sound transport”, the authors realize higher-order topological boundary modes transportation by employing synthetic dimensions. More specifically, the synthetic dimension attributes to the existence of the topological boundary mode and varying the synthetic coordinate along z direction induces boundary modes transport, alike to the well-known Thouless pumping. The physics behind is solid, experimental results are also present to support the simulation results.

This work is interesting in sound wave manipulation, while before I get the conclusion to support its publication in Nature Communication, I would like the authors to address the following remarks and comments.

Response: We thank the reviewer for his/her account of our paper. Below we do our best to answer his/her concerns and suggestions.

1) The author claims the system obeys Eq. (5) in the main text and said “hence the amplitudes of the modes along the z direction are all equal”, which means there would be no reflection when wave propagates along z direction. While, as I can imagine, in extreme case when ϕ is fixed, since the system is periodic instead of be a straight waveguide along z direction, reflection induced by multiple scattering is expected. More simulation results to show the reflectivity from the system is appreciated. How would the reflection affect the outcome of the results?

Response: The reviewer raises an important question. We point out that all our simulations are performed with hard boundary conditions at the top and bottom (along the z-axis) of the acoustic domain. As such, the reflection effects in questions are already incorporated in both our simulations and experiments. On the other hand, the derivation of Eq. (5) assumes an infinitely long system in the z-direction, or a situation where the source and the measurements are performed far away from the top/bottom boundaries. The analysis, however, can be straightforwardly adapted to the case of a finite sample, where the resonant modes satisfying the appropriate boundary conditions can be found as linear superpositions of the $+k_z$ and $-k_z$ solutions of Eq. (5) at certain quantized values of the wave numbers. Therefore, in response to this comment, we modified the text to, first, alert the reader that the analysis starting before Eq. (1) assumes an infinite system and, second, to inform the reader how to adapt the analysis to finite samples. Specifically, in the revised manuscript, we have added the following sentence to the end of Section II-A:

“Last, it is worth mentioning that if the acoustic crystal is finite in the z-direction, the collective resonant modes are given by linear superpositions of the $\pm k_z$ solutions of Eq. (5) and they occur at quantized values of the wave numbers for which the top and bottom boundary conditions are simultaneously satisfied. Since the effective Hamiltonian from Eq. (4) is independent of the sign of the wave number, the main conclusion regarding the horizontal spatial profiles of the modes holds without modifications.”

2) This is a question related to the first one. I note that the total propagating distance along z direction is not long compared to the previous study (Ref [32]), while “adiabatic pumping” requires a slow variation of the parameters. It is important to show that the adiabatic condition is indeed satisfied in this setup. Can the authors provide a quantitative study to support the “adiabatic” claim?

Response: We thank the reviewer for the suggestion and agree that “adiabatic pumping” requires a slow variation of the phase parameters. We point out that the total phason variation from one end to other is just 0.4π and not the whole 2π . For this reason, we are well within the adiabatic regime and a supporting quantitative study to assess the adiabatic limit is included in new Section S2 (“DISCUSSION ON ADIABATICITY”) of the revised Supplementary Information and summarized here briefly:

“In order to assess the adiabatic regime, we calculate the ratio of the pressure amplitude between input A and output B with different numbers of horizontal layers; see Fig. R1a. It is observed that for topological pumping the energy is reliably transported from the left edge to the right edge when $L_z > 10a_z$; see Fig. R1b. In contrast, when $L_z < 10a_z$, the amplitudes of the waveguide drop. Then the adiabatic condition may fail and the reflection between different layers will affect the topological waveguide. Further, we present simulation results of the edge-to-edge and corner-to-corner pumping in Figs. R2 and R3, respectively. It is also observed that our 3D system with $L_z = 15a_z$ has a reliable sound transport.”

Fig. R1: Assessing adiabaticity of the 1D topological pumping. (a) Schematic of the 2D channel-modulated acoustic crystal with input A and output B. (b) Ratio of the pressure amplitude between A and B with different numbers of horizontal layers. The pressure is imposed at A and collected at B.

Fig. R2: Assessing adiabaticity of the edge-to-edge topological pumping. (a) Schematic of the 3D channel-modulated acoustic crystal with input A and output B. (b) Ratio of the pressure amplitude with different numbers of horizontal layers. The pressure is imposed at A and collected at B.

Fig. R3: Assessing adiabaticity of the corner-to-corner topological pumping. (a) Schematic of the 3D channel-modulated acoustic crystal with input A and output B. (b) Ratio of the pressure amplitude with different numbers of horizontal layers. The pressure is imposed at A and collected at B.

3) Could you please figure out the pressure distribution along z direction in boundaries that wave localize?

Response: In the manuscript, Figs. 3b and 3d only show the pressure distribution of edge-bulk-edge and corner-bulk-corner pumping, layer by layer along the z direction, respectively. Following the suggestion of the reviewer, we show below the corresponding pressure distribution along z direction in boundaries; see Fig. R4. We have added this figure in the revised Supplementary Information.

Fig. R4: Topological pumping in the 3D channel-modulated acoustic crystal. (a,b) Pressure distribution of edge-bulk-edge pumping along z direction in boundaries. **(c,d)** Pressure distribution of corner-bulk-corner pumping along z direction in boundaries.

4) The author shows the system is a tight-binding system and presents the fitting results in Supplemental Fig. S1. My puzzle is, since the system is a 3x3 model, why only two bands are presented? Why specific phi is selected?

Response: We thank the reviewer for pointing out this discrepancy. In Supplementary Fig. S1, we only show two highest bands of the TB model for 1D pumping (Fig. S1a) and 2D pumping (Fig. S1b). Indeed, the 3×1 model (Fig. S1a) has three bulk bands separated by two band gaps; the 3×3 model (Fig. S1b) has five bulk band regions separated by four band gaps. For clarity, we have modified Fig. S2b and Fig. S3b to make appear fitting results of all bands at once with $\phi \in [-\pi, \pi]$; see Fig. R5.

Fig. R5: The fitting results of the TB model for 1D pumping (a) and 2D pumping (b) when the pumping parameter is $\phi = -0.2\pi$. The grey surfaces are the full-wave simulation results. The corresponding fitting results are in red circles. The fitting parameters for 1D pumping are $\varepsilon = 3.61$, $\kappa_0 = -1.44$ and $\delta = 0.55$. The fitting parameters for 2D pumping are $\varepsilon = 4.43$, $\kappa_0 = -0.91$ and $\delta = 0.55$. The first and second Chern numbers are assigned to each band gap in (a) and (b), respectively.

5) Sound waves of 7 KHz in 3D printing structures may suffer considerable loss, which will broaden the spectrum, making the boundary mode and bulk mode overlap. Discussion about the loss is absent in the whole article.

Response: Dissipation is indeed a factor that needs to be carefully assessed for acoustic crystals. From the beginning, our experimental design focused on minimizing these effects, by relying on the state of the art in the 3D printing and by having the crystal as a solid one single piece rather than assembled from individual resonators. Based on the negligible signal attenuation in measurements, we concluded that, for the system sizes probed in our experiments, dissipation can be ignored. Still, the reviewer raises a valid point, namely, are the topological gaps large enough relative to the broadening due to the dissipation? The answer is yes because the modulation of the channels was adjusted until the topological gaps opened substantially ($\approx 1\text{kHz}$, see Fig. S1).

To further demonstrate this point, we evaluate the loss effects on topological boundary modes by introducing loss in the hopping $\kappa_j(\phi) = \kappa_0[1 + \delta \cos(b_j + \phi)] + i\eta$, where η is the loss coefficient. Figure R6 shows the edge-state dispersion of the 2D system (Fig. 1a) with different loss coefficients. In Fig. R6a, it is clearly evident that the smaller loss coefficient with $\eta = 0.1\kappa_0$ cannot affect the dispersion property by comparing with the lossless case in Fig. S2d. However, as seen in Fig. R6b, the larger loss seriously affected the dispersion property, which makes the boundary mode and bulk mode overlap. This phenomenon is further confirmed in the 3D system; see Fig. R7. In the revised Supplementary Information, we have augmented a new Section S1-D (“Modulated channels with loss”) to include detailed discussions regarding loss effects.

Fig. R6: The edge-state dispersion of the 2D channel-modulated acoustic crystal with different losses in the hopping. (a) $\eta = \kappa_0$. (b) $\eta = 6\kappa_0$.

Fig. R7: The edge- and corner-state dispersion of the 3D channel-modulated acoustic crystal with different losses in the hopping. (a) $\eta = \kappa_0$. (b) $\eta = 3\kappa_0$.

In addition, as a response to this point raised by the reviewer, we have added the following sentence to Methods (“*Experimental specification*”) of the revised manuscript:

“The dissipation effects are minimized by employing the state of the art in 3D printing, which delivers acoustic crystals made of a single solid piece of high-quality polymer. Furthermore, the modulation of the connecting channels is adjusted until the topological gaps opened appreciably such that the dynamics of the topological pumped modes is virtually unaffected by dissipation effects, at least for the system sizes considered in our studies.”

6) Two bands with nonzero second Chern number are presented in Fig. S3b, what about other bands and what is the relation between second Chern number and the corner mode?

Response: We thank the reviewer for the above comments.

- (1) The second Chern numbers for the four band gaps in Fig. S3b are 1, -3, 3 and -1, respectively; see Fig. R5b. Two missing second gap Chern numbers have now been assigned to the corresponding band gaps in Supplementary Fig. S3b.
- (2) Indeed, the effective Hamiltonian of our 3D system is a sum of two copies of the Hamiltonian from our 2D system, $H(\phi) = H(\phi^x) \otimes I + I \otimes H(\phi^y)$, which implies that the corner mode are composed of the product of two edge modes in the 2D system (Zilberberg *et al.*, *Nature* 553, 59-62, 2018). As a result, the topological corner modes in the 3D system are directly related to the nonzero combinations of two first gap Chern numbers, $C \equiv (C_1^x, C_1^y)$, which are further connected to the second gap Chern number. It reads (Chen *et al.*, *Phys. Rev. X* 11, 011016, 2021)

$$\sum_{i=1}^r C_{2,f_i} = \sum_{f_{r-1} < f_x + f_y < f_r} C_{1,f_x}^x C_{1,f_y}^y,$$

where C_{2,f_i} is the second Chern number for the band gap with the frequency no larger than f_i , and C_{1,f_x}^x is the first Chern number for the band gap with the frequency no larger than f_x . In our system, the topological invariants protecting the corner modes from the lowest to the highest band gaps are $(C_{1,f_1}^x, C_{1,f_1}^y) = (-1, -1)$, $(C_{1,f_1}^x, C_{1,f_2}^y) = (-1, 1)$, $(C_{1,f_2}^x, C_{1,f_1}^y) = (1, -1)$, and $(C_{1,f_2}^x, C_{1,f_2}^y) = (1, 1)$. The above discussion and the related references have been added to the end of Section S1-C of the revised Supplementary Information.

7) Color bars for the eigenstate field distributions in all figures are absent.

Response: We thank the reviewer for revealing this shortcoming: color bars have been added to Figs. 1d, 2d, and S9-S12 to clarify the scale of intensity.

Reviewer #2 (Remarks to the Author):

In this work, 1D and 2D topological boundary pumping is demonstrated using arrays of coupled acoustic cavities. Effectively, linear sound propagation is controlled by a careful design of the array structure and coupling to topological boundary modes is done using a speaker [akin to previous works in the field: Nature Physics volume 15, pages357–361(2019)]. The chosen metamaterial structures and experimental ideas/observables reproduce the results from Refs. [23] and [20] for acoustic devices. The robustness of the 2D pumping to disorder in the bulk is further explored. The extension of topological boundary pumping is important to for potential applications. Connection to high-order topological insulators is commented on.

The exploration of topological effects in driven-dissipative metamaterials has started in the optics domain and since sparked immense activity in other linear media. The present results similarly provide a comprehensive transition of topological boundary pumping results from optics to acoustics. As such, they provide a novel platform for harnessing such effects for applications. I would, therefore, recommend publication in Nat. Commun. after the following points are discussed:

Response: We thank the reviewer for his/her account of our paper. Below we do our best to answer his/her concerns and suggestions.

1. The topology of pumps relies on the adiabatic theorem. At the same time, in the experiment adiabaticity is broken. Nevertheless, boundary pumping is observed. I would like the authors to discuss this point, see, e.g., Nature Communications volume 11, 3758 (2020).

Response: We thank the reviewer for pointing us the interesting research from Nature Communications volume 11, 3758, 2020. The non-Hermitian strategy employed there could be used to reduce the size of our topological acoustic pumps, therefore, we have added the following sentences to the end of Section III of the revised manuscript:

“At last, we emphasize that, in order to achieve a reasonable adiabatic regime, the number of stacks in our experimental set-ups is appreciable and, while this is perfectly fine for demonstration purposes, it will be an obstacle for practical applications. Recently, Fedorova et al (Nature Communications volume 11, 3758, 2020) showed that the non-adiabatic effects can be compensated using modulated dissipative channels and that such strategy can be used to achieve quantized topological pumping with fast driven cycles. It will be interesting to explore if this strategy can be deployed for our acoustic crystals in order to reduce the number of stacks needed for the topological pumping of sound.”

We would like to include a few further comments for the reviewer. First, note that our system is not driven in an adiabatic cycle. Instead, an open (as opposed to closed) path was selected in the phason space, such that at the beginning/end of the path we have resonant edge modes located at the left/right sides of the structures. This path is slowly “walked” by changing the phason along

the vertical stacking. The second remark is that the z -coordinate is discrete in our case. As such, the connection with the adiabatic theorem cannot be made. The theory we developed looks directly at the spatial profile of the resonant modes excited at a fixed frequency (in a controlled ϵ -expansion in Eq. (5)). The edge-to-edge pumping, which is predicted and then observed, is simply an interplay between the wave-dispersions along the stackings and on the horizontal planes. As long as ϵ remains within reasonable bounds, the predicted spatial profiles should be close to reality. In the revised Supplementary Information, we have augmented a new section to include a quantitative assessment of the “non-adiabatic” effects, with the conclusion that our ϵ is well inside the asymptotic regime where the observations become independent of this parameter. See also our response to Comment (5) from Reviewer #1.

2. As the present study is mostly focused on transferring ideas from other fields for the purpose of novel applications, I would like the authors to provide more details on the proposed metrology applications.

Response: We thank the reviewer for the suggestion and agree that the details on the proposed metrology applications merits further discussion. In our opinion, the modulated acoustic crystal with synthetic spaces offers a new platform and route for efficient acoustic topological mode transport through the phason engineering. This may exhibit the possibility to implement the topological split-flow device, such as the topological beam splitter. In Fig. R8a, we create an acoustic beam splitter to engineer two-way beam splitting. In the current design, we consider a 2D system with different phason orbits in four quadrants: in the first quadrant, the phason value is linearly distributed from -0.2π to 0.2π along the z -direction; in the second quadrant, the phason value is linearly distributed from 0.2π to -0.2π along the z -direction; in the third quadrant, the phason value is held constant with $\phi = 0.2\pi$ along the z -direction; in the fourth quadrant, the phason value is held constant with $\phi = -0.2\pi$ along the z -direction. The input point is located at the bottom edge between the third and fourth quadrants. As shown in Fig. R8b, the sound stays confined to the interface until it arrives at the junction of four quadrants. Then, it splits into the first and second quadrants, and eventually reaches the two-end sites. Thanks to the topological protection, the propagation is immune against back reflection from discontinuity. As such, our design, based on the phason engineering and the topological pumping, provides a new avenue on the application of acoustic beam splitters. In Section III of the revised manuscript, we have included the related discussion on the application of acoustic beam splitters.

Fig. R8: Topologically protected acoustic beam splitter. (a) Schematic of the phason engineering in the 2D acoustic system. (b) The pressure field distribution of the beam splitter. The sound is injected at the center of the bottom edge with the frequency at $f = 4960$ Hz.

REVIEWERS' COMMENTS

Reviewer #1 (Remarks to the Author):

The authors have clarified the issues raised in the last round. In light of the revised manuscript, I think the authors need to add Phys. Rev. X 11, 011016, 2021 to the reference in the main text. Another recent work on acoustic topological pumping (PRL 126, 054301,2021) is highly relevant to this work. It also needs to be cited and discussed.

Reviewer #2 (Remarks to the Author):

The authors diligently answered the critique raised by the referees and improved the quality of their manuscript. I support publication of the work in Nature Comm.

Response to the Reviewers' Comments

We thank all reviewers for their efforts and support to publish the manuscript in Nature Communications.

Reviewer #1 (Remarks to the Author):

The authors have clarified the issues raised in the last round. In light of the revised manuscript, I think the authors need to add Phys. Rev. X 11, 011016, 2021 to the reference in the main text. Another recent work on acoustic topological pumping (PRL 126, 054301,2021) is highly relevant to this work. It also needs to be cited and discussed.

Response: We thank the reviewer for the suggestion. The relevant references have been added to the revised manuscript.

Reviewer #2 (Remarks to the Author):

The authors diligently answered the critique raised by the referees and improved the quality of their manuscript. I support publication of the work in Nature Comm.